# Motif-based Graph Self-Supervised Learning for Molecular Property Prediction

**Zaixi Zhang**[1], **Qi Liu**[1*], **Hao Wang**[1], **Chengqiang Lu**[1], **Chee-Kong Lee**[2*]
1: Anhui Province Key Lab of Big Data Analysis and Application,
School of Computer Science and Technology,
University of Science and Technology of China
2: Tencent America
{zaixi, wanghao3, lunar}@mail.ustc.edu.cn, qiliuql@ustc.edu.cn
cheekonglee@tencent.com

## Abstract

Predicting molecular properties with data-driven methods has drawn much attention in recent years. Particularly, Graph Neural Networks (GNNs) have demonstrated remarkable success in various molecular generation and prediction tasks. In cases where labeled data is scarce, GNNs can be pre-trained on unlabeled molecular data to first learn the general semantic and structural information before being finetuned for specific tasks. However, most existing self-supervised pre-training frameworks for GNNs only focus on node-level or graph-level tasks. These approaches cannot capture the rich information in subgraphs or graph motifs. For example, functional groups (frequently-occurred subgraphs in molecular graphs) often carry indicative information about the molecular properties. To bridge this gap, we propose Motif-based Graph Self-supervised Learning (MGSSL) by introducing a novel self-supervised motif generation framework for GNNs. First, for motif extraction from molecular graphs, we design a molecule fragmentation method that leverages a retrosynthesis-based algorithm BRICS and additional rules for controlling the size of motif vocabulary. Second, we design a general motif-based generative pre-training framework in which GNNs are asked to make topological and label predictions. This generative framework can be implemented in two different ways, i.e., breadth-first or depth-first. Finally, to take the multi-scale information in molecular graphs into consideration, we introduce a multi-level self-supervised pre-training. Extensive experiments on various downstream benchmark tasks show that our methods outperform all state-of-the-art baselines.

## 1 Introduction

The past decade has witnessed the remarkable success of deep learning in natural language processing (NLP) [5], computer vision (CV) [13], and graph analysis [19]. Inspired by these developments, researchers in the chemistry domain have also tried to exploit deep learning methods for molecule-based tasks such as retrosynthesis [47] and drug discovery [8]. To preserve the internal structural information, molecules can be naturally modeled as graphs where nodes represent atoms and edges denote the chemical bonds. Recently, some works applied Graph Neural Network (GNN) and some of its variants for molecular property prediction and obtained promising results [8, 25].

Though GNNs have achieved remarkable accuracy on molecular property prediction, they are usually data-hungry, i.e. a large amount of labeled data (i.e., molecules with known property data) is required for training [8, 12]. However, labeled molecules only occupy an extremely small portion of the

---

*Qi Liu and Chee-Kong Lee are corresponding authors.

35th Conference on Neural Information Processing Systems (NeurIPS 2021).

enormous chemical space since they can only be obtained from wet-lab experiments or quantum chemistry calculations, which are time-consuming and expensive. Moreover, directly training GNNs on small labeled molecule datasets in a supervised fashion is prone to over-fitting and the trained GNNs can hardly generalize to out-of-distribution data.

Similar issues have also been encountered in natural language processing and computer vision. Recent advances in NLP and CV address them by self-supervised learning (SSL) where a model is first pre-trained on a large unlabeled dataset and then transferred to downstream tasks with limited labels [13, 2, 5]. For example, the pre-trained BERT language model [5] is able to learn expressive contextualized word representations through reconstructing the input text-next sentence and masked language predictions so that it can significantly improve the performance of downstream tasks. Inspired by these developments, various self-supervised pre-training methods of GNNs have been proposed [14, 15, 52, 51, 32, 10, 30]. Based on how the self-supervised tasks are constructed, these methods can be classified into two categories, contrastive methods and predictive methods. Contrastive methods force views from the same graph (e.g., sampling nodes and edges from graphs) to become closer and push views from different graphs apart. On the other hand, the predictive methods construct prediction tasks by exploiting the intrinsic properties of data. For example, Hu et.al. [14] designed node-level pre-training tasks such as predicting the context of atoms and the attributes of masked atoms and bonds. [15] introduced an attributed graph reconstruction task where the generative model predicts the node attributes and edges to be generated at each step.

However, we argue that existing self-supervised learning tasks on GNNs are sub-optimal since most of them fail to exploit the rich semantic information from graph motifs. Graph motifs can be defined as significant subgraph patterns that frequently occur [27]. Motifs usually contain semantic meanings and are indicative of the characteristics of the whole graph. For example, the hydroxide (–OH) functional group in small molecules typically implies higher water solubility. Therefore, it is vital to design motif-level self-supervised learning tasks which can benefit downstream tasks such as molecular property prediction.

Designing motif-level self-supervised learning tasks brings unique challenges. First, existing motif mining techniques could not be directly utilized to derive expressive motifs for molecular graphs because they only rely on the discrete count of subgraph structures and overlook the chemical validity [1, 18]. Second, most graph generation techniques generate graphs node-by-node [23, 50], which are not suitable for our task of motif generation. Finally, how to unify multi-level self-supervised pre-training tasks harmoniously brings a new challenge. One naive solution to do pre-training tasks sequentially may lead to catastrophic forgetting similar to continual learning [7].

To tackle the aforementioned challenges, we propose **M**otif-based **G**raph **S**elf-**S**upervised **L**earning (MGSSL) and Multi-level self-supervised pre-training in this paper. Firstly, MGSSL introduces a novel motif generation task that empowers GNNs to capture the rich structural and semantic information from graph motifs. To derive semantically meaningful motifs and construct motif trees for molecular graphs, we leverage the BRICS algorithm [4] which is based on retrosynthesis from the chemistry domain. Two additional fragmentation rules are further introduced to reduce the redundancy of motif vocabulary. Secondly, a general motif-based generative pre-training framework is designed to generate molecular graphs motif-by-motif. The pre-trained model is required to make topology and attribute predictions at each step and two specific generation orders are implemented (breadth-first and depth-first). Furthermore, to take the multi-scale regularities of molecules into consideration, we introduce Multi-level self-supervised pre-training for GNNs where the weights of different SSL tasks are adaptively adjusted by the Frank-Wolfe algorithm [16]. Finally, by pre-training GNNs on the ZINC dataset with our methods, the pre-trained GNNs outperforms all the state-of-the-art baselines on various downstream benchmark tasks, demonstrating the effectiveness of our design. The implementation is publicly available at https://github.com/zaixizhang/MGSSL.

## 2 Preliminaries and Related Work

### 2.1 Molecular Property Prediction

Prediction of molecular properties is a central research topic in physics, chemistry, and materials science [44]. Among the traditional methods, density functional theory (DFT) is the most popular one and plays a vital role in advancing the field [21]. However, DFT is very time-consuming and its complexity could be approximated as $\mathcal{O}(N^3)$, where $N$ denotes the number of particles. To find

more efficient solutions for molecular property prediction, various machine learning methods have been leveraged such as kernel ridge regression, random forest, and convolutional neural networks [6, 26, 36]. To fully consider the internal spatial and distance information of atoms in molecules, many works [8, 35, 48, 25] regard molecules as graphs and explore the graph convolutional network for property prediction. To better capture the interactions among atoms, [8] proposes a message passing framework and [20, 48] extend this framework to model bond interactions. [25] builds a hierarchical GNN to capture multilevel interactions. Furthermore, [32] integrates GNNs into the Transformer-style architecture to deliver a more expressive model.

## 2.2 Preliminaries of Graph Neural Networks

Recent years have witnessed the success of Graph Neural Networks (GNNs) for modeling graph data [11, 19, 40, 46, 43, 42, 55]. Let $G = (V, E)$ denotes a graph with node attributes $X_v$ for $v \in V$ and edge attributes $e_{uv}$ for $(u, v) \in E$. GNNs leverage the graph connectivity as well as node and edge features to learn a representation vector (i.e., embedding) $h_v$ for each node $v \in G$ and a vector $h_G$ for the entire graph $G$. Generally, GNNs follows a message passing paradigm, in which representation of node $v$ is iteratively updated by aggregating the representations of $v$'s neighboring nodes and edges. Specifically, there are two basic operators for GNNs: AGGREGATE(·) and COMBINE(·). AGGREGATE(·) extracts the neighboring information of node $v$; COMBINE(·) serves as the aggregation function of the neighborhood information. After $k$ iterations of aggregation, $v$'s representation captures the structural information within its $k$-hop network neighborhood. Formally, the $k$-th layer of a GNN is:

$$h_v^{(k)} = \text{COMBINE}^{(k)}\left(h_v^{(k-1)}, \text{AGGREGATE}^{(k)}\left(\left\{(h_v^{(k-1)}, h_u^{(k-1)}, e_{uv}) : u \in \mathcal{N}(v)\right\}\right)\right), \quad (1)$$

where $h_v^{(k)}$ denotes the representation of node $v$ at the $k$-th layer, $e_{uv}$ is the feature vector of edge between $u$ and $v$ and and $\mathcal{N}(v)$ represents the neighborhood set of node $v$. $h_v^{(0)}$ is initialized with $X_v$. Furthermore, to obtain the representation of the entire graph $h_G$, READOUT functions are designed to pool node representations at the final iteration $K$,

$$h_G = \text{READOUT}\left(\left\{h_v^{(K)} | v \in G\right\}\right). \quad (2)$$

READOUT is a permutation-invariant function, such as averaging, sum, max or more sophisticated graph-level pooling functions [49, 53].

## 2.3 Self-supervised Learning of Graphs

Graph Self-supervised learning aims to learn the intermediate representations of unlabeled graph data that are useful for unknown downstream tasks. [14, 15, 52, 51, 32, 10, 30, 39, 28, 41]. Traditional graph embedding methods [9, 29] define different kinds of graph proximities, i.e., the vertex proximity, as the self-supervised objective to learn vertex representations. Furthermore, [39, 28, 41] proposed to use the mutual information maximization as the optimization objective for GNNs. Recently, more self-supervised tasks for GNNs have been proposed [14, 15, 52, 51, 32, 10, 54, 30]. Based on how the self-supervised tasks are constructed, these models can be classified into two categories, namely contrastive models and predictive models. Contrastive models try to generate informative and diverse views from data instances and perform node-to-context [14], node-to-graph [39] or motif-to-graph [54] contrastive learning. On the other hand, predictive models are trained in a supervised fashion, where the labels are generated based on certain properties of the input graph data, i.e., node attributes [32], or by selecting certain parts of the graph [15, 32].

However, few works try to leverage the information of graph motifs for graph self-supervised learning. Grover [32] use traditional softwares to extract motifs and treat them as classification labels. MICRO-Graph [54] exploits graph motifs for motif-graph contrastive learning. Unfortunately, both methods fail to consider the topology information of motifs.

## 3 Motif-based Graph Self-supervised Learning

In this section, we introduce the framework of motif-based graph self-supervised learning (Figure 1). Generally, the framework consists of three parts: chemistry-inspired molecule fragmentation, motif generation and multi-level self-supervised pre-training for GNNs.

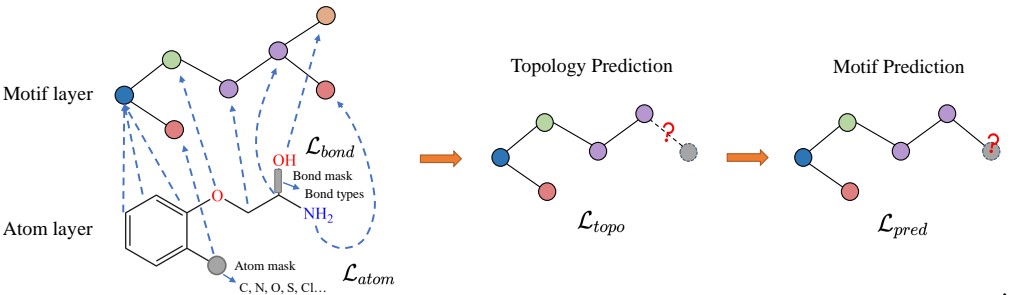

Figure 1: Illustration of Motif-based Graph Self-supervised learning (MGSSL). The multi-level pre-training consists of two layers, Atom layer and Motif layer. In the Atom layer, we mask node/edge attributes and let GNNs predict those attributes based on neighboring structures. In the Motif layer, we construct motif trees and perform motif generative pre-training. In each step, based on existing motifs and connections, topology and motif predictions are made iteratively.

### 3.1   Chemistry-inspired Molecule Fragmentation

Given a dataset of molecules, the first step of our method is to decompose molecules into several fragments/motifs. Based on these fragments, a molecule graph can be converted into a motif tree structure where each node represents a motif and the edges denotes the relative spatial relationships among motifs. We choose to represent the structure of motifs as a tree because it facilitates the motif generation task. Formally, given a molecule graph $G = (V, E)$, a motif tree $\mathcal{T}(G) = (\mathcal{V}, \mathcal{E}, \mathcal{X})$ is a connected labeled tree whose node set is $\mathcal{V} = \{M_1, ..., M_n\}$ and edge set is $\mathcal{E}$. $\mathcal{X}$ refers to the induced motif vocabulary. Each motif $M_i = (V_i, E_i)$ is a subgraph of $G$. There are many ways to fragment a given graph while the designed molecule fragmentation method should achieve the following goals: 1) In a motif tree $\mathcal{T}(G)$, the union of all motifs $M_i$ should equals $G$. Formally, $\bigcup_i V_i = V$ and $\bigcup_i E_i \bigcup \mathcal{E} = E$. 2) In a motif tree $\mathcal{T}(G)$, the motifs should have no intersections. That is $M_i \cap M_j = \emptyset$. 3) The induced motifs should capture semantic meanings, e.g., similar to meaningful functional groups in the chemistry domain. 4) The occurrence of motifs in the dataset should be frequent enough so that the pre-trained GNNs can learn semantic information of motifs that can be generalized to downstream tasks. After the molecule fragmentation, the motif vocabulary of the molecule dataset should have a moderate size.

In Figure 2, we show the overview of molecule fragmentation. Generally, there are three procedures, BRICS fragmentation, further decomposition, and motif tree construction. A motif vocabulary can be built via preprocessing the whole molecule dataset following the molecule fragmentation procedures.

To fragment molecule graphs and construct motif trees, we firstly use the Breaking of Retrosynthetically Interesting Chemical Substructures (BRICS) algorithm [4] that leverages the domain knowledge from chemistry. BRICS defines 16 rules and breaks strategic bonds in a molecule that match a set of chemical reactions. "Dummy" atoms are attached to each end of the cleavage sites, marking the location where two fragments can join together. BRICS cleavage rules are designed to retain molecular components with valuable structural and functional content, e.g. aromatic rings.

However, we find BRICS alone cannot generate desired motifs for molecule graphs. This is because BRICS only breaks bonds based on a limited set of chemical reactions and tends to generate several large fragments for a molecule. Moreover, due to the combination explosion of graph structure, we find BRICS produces many motifs that are variations of the same underlying structure (e.g., Furan ring with different combinations of Halogen atoms). The motif vocabulary is large (more than 100k unique fragments) while most of these motifs appear less than 5 times in the whole dataset.

To tackle the problems above, we introduce a post-processing procedure to BRICS. To alleviate the combination explosion, we define two rules operating on the output fragments of BRICS: (1) Break the bond where one end atom is in a ring while the other end not. (2) Select non-ring atoms with three or more neighboring atoms as new motifs and break the neighboring bonds. The first rule reduces the number of ring variants and the second rule break the side chains. Experiments show that these rules effectively reduce the size of motif vocabulary and improve the occurrence frequency of motifs.

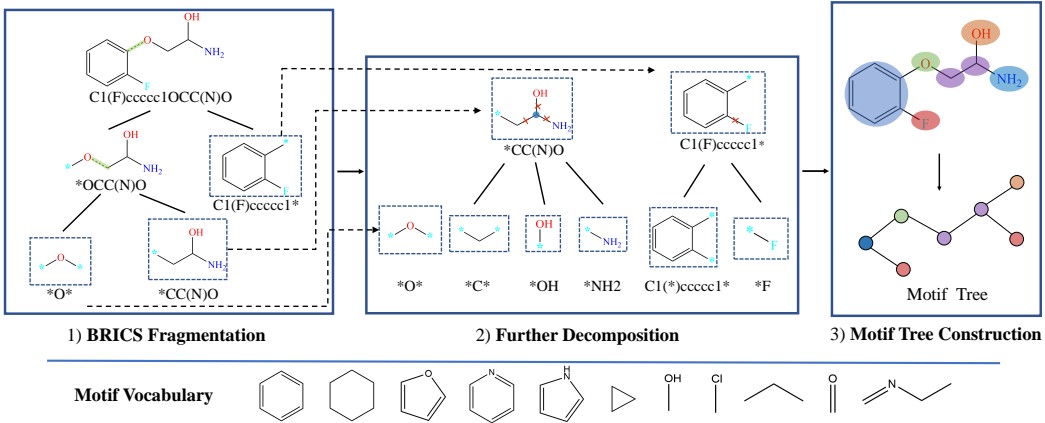

Figure 2: Overview of molecule fragmentation. Generally, there are three steps: 1) Firstly a molecule graph is cleaved based on BRICS. 2) Further decomposition to reduce the redundancy of motifs 3) Construct motif trees from molecule graphs. A motif vocabulary is built after preprocessing the whole molecule dataset.

## 3.2 Motif Generation

Here, we present the framework of motif generation for the generative self-supervised pre-training. The goal of the motif generation task is to let GNNs learn the data distribution of graph motifs so that the pre-trained GNNs can easily generalize to downstream tasks with several finetuning steps on graphs from similar domains.

Given a molecule graph $G = (V, E)$ and a GNN model $f_\theta$, we first convert the molecule graph to a motif tree $\mathcal{T}(G) = (\mathcal{V}, \mathcal{E}, \mathcal{X})$. Then we can model the likelihood over this motif tree by the GNN model as $p(\mathcal{T}(G); \theta)$, representing how the motifs are labeled and connected. Generally, our method aims to pretrain the GNN model $f_\theta$ via maximizing the likelihood of motif trees, i.e., $\theta^* = \arg\max_\theta p(\mathcal{T}(G); \theta)$. To model the likelihood of motif trees, special predictions heads for topology and motif label predictions are designed (as shown in the following sections) and optimized along with $f_\theta$. After pre-training, only the GNN model $f_\theta$ is transferred to downstream tasks.

We note that most existing works on graph generation [17, 23] follow the auto-regressive manner to factorize the probability objective, i.e., $p(\mathcal{T}(G); \theta)$ in this work. For each molecular graph, they decompose it into a sequence of generation steps. Similarly in this paper, we interleave the addition of a new motif, and the addition of bonds to connect the newly added motif to the existing partial motif tree. We denote a permutation vector $\pi$ to determine the motif ordering, in which $i^\pi$ denotes the motif ID of $i$-th position in permutation $\pi$. Therefore, the probability $p(\mathcal{T}(G); \theta)$ is equivalent to the expected likelihood over all possible permutations, i.e.,

$$p(\mathcal{T}(G); \theta) = \mathbb{E}_\pi \left[ p_\theta(\mathcal{V}^\pi, \mathcal{E}^\pi) \right], \qquad (3)$$

where $\mathcal{V}^\pi$ denotes the permuted motif labels and $\mathcal{E}^\pi$ denotes the edges among motifs.

Our formalism permits a variety of orders. For simplicity, we assume that any node ordering $\pi$ has an equal probability and we also omit the subscript $\pi$ when illustrating the generative process for one permutation in the following sections. Given a permutation order, the probability of generating motif tree $\mathcal{T}(G)$ can be decomposed as follows:

$$\log p_\theta(\mathcal{V}, \mathcal{E}) = \sum_{i=1}^{|\mathcal{V}|} \log p_\theta(\mathcal{V}_i, \mathcal{E}_i \mid \mathcal{V}_{<i}, \mathcal{E}_{<i}). \qquad (4)$$

At each step $i$, we use motif attributes $\mathcal{V}_{<i}$ and structures $\mathcal{E}_{<i}$ of all motifs generated before $i$ to generate a new motif $\mathcal{V}_i$ and its connection with existing motifs $\mathcal{E}_i$.

Equation 4 describes the autoregressive generative process of motif trees. Then the question is how to choose an efficient generation order and how to model the conditional probability $\log p_\theta(\mathcal{V}_i, \mathcal{E}_i \mid \mathcal{V}_{<i}, \mathcal{E}_{<i})$. In the following sections, we introduce two efficient generation orders,

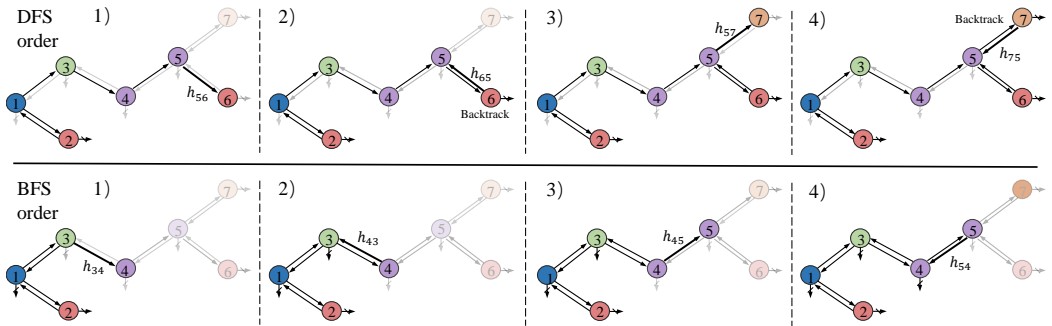

Figure 3: Illustration of the motif generation orders. The first row is the illustration of DFS order and the second row is the BFS order.

breadth-first (BFS) and depth-first (DFS) orders and show the corresponding auto-regressive generative models.

To generate a motif tree from scratch, we need to first choose the root of motif tree. In our experiments, we simply choose the motif with the first atom in the canonical order [34]. Then MGSSL generates motifs in DFS or BFS orders (see Figure 3). In the DFS order, for every visited motif, MGSSL first makes a topological prediction: whether this node has children to be generated. If a new child motif node is generated, we predict its label and recurse this process. MGSSL backtracks when there is no more children to generate. As for the BFS order, MGSSL generates motif nodes layer-wise. For motifs nodes in the $k$-th layer, MGSSL makes topological predictions and label predictions. If all the children nodes of motifs in the $k$-th layer are generated, MGSSL will move to the next layer. We also note that the motif node ordering in BFS and DFS are not unique as the order within sibling nodes is ambiguous. In the experiments, we pre-train in one order and leave this issue and other potential generation orders for future exploration.

At each time step, a motif node receives information from other generated motifs for making those predictions. The information is propagated through message vectors $h_{ij}$ when motif trees are incrementally constructed. Formally, let $\hat{\mathcal{E}}_t$ be the set of message at time $t$. The model visits motif $i$ at time $t$ and $x_i$ denotes the embedding of motif $i$, which can be obtained by pooling the atom embeddings in motif $i$. The message $\mathrm{h}_{i,j}$ is updated through previous messages:

$$\mathrm{h}_{i,j} = \mathrm{GRU}\left(x_i, \ \{\mathrm{h}_{k,i}\}_{(k,i)\in\hat{\mathcal{E}}_t, k\neq j}\right), \tag{5}$$

where GRU is Gated Recurrent Unit [3] adapted for motif tree message passing:

$$s_{i,j} = \sum_{(k,i)\in\hat{\mathcal{E}}_t, k\neq j} \mathrm{h}_{k,i} \tag{6}$$

$$z_{i,j} = \sigma(\mathrm{W}^z x_i + \mathrm{U}^z s_{i,j} + b^z) \tag{7}$$

$$r_{k,i} = \sigma(\mathrm{W}^r x_i + \mathrm{U}^r \mathrm{h}_{k,i} + b^r) \tag{8}$$

$$\tilde{\mathrm{h}}_{i,j} = \tanh(\mathrm{W} x_i + U \sum_{k=\mathcal{N}(i)\setminus j} r_{k,i} \odot \mathrm{h}_{k,i}) \tag{9}$$

$$\mathrm{h}_{i,j} = (1 - z_{ij}) \odot s_{ij} + z_{ij} \odot \tilde{\mathrm{h}}_{i,j}. \tag{10}$$

**Topology Prediction:** When MGSSL visits motif $i$, it needs to make binary predictions on whether it has children to be generated. We compute the probability via a one hidden layer network followed by a sigmoid function taking messages and motif embeddings into consideration:

$$p_t = \sigma\left(U^d \cdot \tau(W_1^d x_i + W_2^d \sum_{(k,i)\in\hat{\mathcal{E}}_t} h_{k,i})\right), \tag{11}$$

where $d$ is the dimension of the hidden layer.

**Motif Label Prediction:** When motif $i$ generate a child motif $j$, we predict the label of child $j$ with:

$$q_j = \mathrm{softmax}(U^l \tau(W^l h_{ij})), \tag{12}$$

where $q_j$ is the distribution over the motif vocabulary $\mathcal{X}$ and $l$ is the hidden layer dimension. Let $\hat{p}_t \in \{0, 1\}$ and $\hat{q}_j$ be the ground truth topological and motif label values, the motif generation loss is the sum of cross-entropy losses of topological and motif label predictions:

$$\mathcal{L}_{motif} = \sum_t \mathcal{L}_{topo}(p_t, \hat{p}_t) + \sum_j \mathcal{L}_{pred}(q_j, \hat{q}_j). \tag{13}$$

In the optimization process, minimizing the above loss function corresponds to maximizing the log likelihood in equation 4. Note that in the training process, after topological and motif label prediction at each step, we replace them with their ground truth so that the MGSSL makes predictions based on correct histories.

### 3.3 Multi-level Self-supervised Pre-training

To capture the multi-scale information in molecules, MGSSL is designed to be a hierarchical framework including Atom-level and Motif-level tasks (Figure 1). For Atom-level pre-training, we leverage attribute masking to let GNNs firstly learn the regularities of the node/edge attributes. In attribute masking, randomly sampled node and bond attributes (e.g., atom numbers, bond types) are replaced with special masked indicators. Then we apply GNNs to obtain the corresponding node/edge embeddings (edge embeddings can be obtained as a combination of node embeddings of the edge's end nodes). Finally, a fully connected layer on top of the embeddings predicts the node/edge attributes. The cross-entropy prediction losses are denoted as $\mathcal{L}_{atom}$ and $\mathcal{L}_{bond}$ respectively.

To avoid catastrophic forgetting in sequential pre-training, we unify multi-level tasks and aim to minimize the hybrid loss in pre-training:

$$\mathcal{L}_{ssl} = \lambda_1 \mathcal{L}_{motif} + \lambda_2 \mathcal{L}_{atom} + \lambda_3 \mathcal{L}_{bond}, \tag{14}$$

where $\lambda_i$ are weights of losses. However, it is time-consuming to do a grid search to determine the optimal weights. Here, we adapts the MGDA-UB algorithm [37] from multi-task learning to efficiently solve the optimization problem (Equation 14). Since MGDA-UB calculates the weights $\lambda_i$ by Frank-Wolfe algorithm [16] at each training step, we do not have to give weights explicitly. The pseudo codes of the training process is included in the Appendix.

## 4 Experimental Results

### 4.1 Experimental Settings

**Datasets and Dataset Splittings.** In this paper, we mainly focus on the molecular property prediction tasks, where large-scale unlabeled molecules are abundant whereas downstream labeled data is scarce. Specifically, we use 250k unlabeled molecules sampled from the ZINC15 database [38] for self-supervised pre-training tasks. As for the downstream finetune tasks, we consider 8 binary classification benchmark datasets contained in MoleculeNet [45]. The detailed dataset statistics are summarized in the Appendix. We use the open-source package RDKit [22] to preprocess the SMILES strings from various datasets. To mimic the real-world use case, we split the downstream dataset by *scaffold-split* [14, 31], which splits the molecules according to their structures. We apply 3 independent runs on random data splitting and report the means and standard deviations.

**Baselines.** We comprehensively evaluate the performance of MGSSL against five state-of-the-art self-supervised pre-training methods for GNNs:

- **Deep Graph Infomax** [41] maximizes the mutual information between the representations of the whole graphs and the representations of its sampled subgraphs.
- **Attribute masking** [14] masks node/edge features and let GNNs predict these attributes.
- **GCC** [30] designs the pretraining task as discriminating ego-networks sampled from a certain node ego-networks sampled from other nodes.
- **Grover** [32] predicts the contextual properties based on atom embeddings to encode contextual information into node embeddings.
- **GPT-GNN** [15] is a generative pretraining task which predicts masked edges and node attributes.

Table 1: Test ROC-AUC (%) performance on molecular property prediction benchmarks using different pre-training strategies with GIN. The rightmost column averages the mean of test performance across the 8 datasets. The best result for each dataset are bolded.

| SSL methods | muv | clintox | sider | hiv | tox21 | bace | toxcast | bbbp | Avg. |
|---|---|---|---|---|---|---|---|---|---|
| No pretrain | 71.7±2.3 | 58.2±2.8 | 57.2±0.7 | 75.4±1.5 | 74.3±0.5 | 70.0±2.5 | 63.3±1.5 | 65.5±1.8 | 67.0 |
| Infomax | 75.1±2.8 | 73.0±3.2 | 58.2±0.5 | 76.5±1.6 | 75.2±0.3 | 75.6±1.0 | 62.8±0.6 | 68.1±1.3 | 70.6 |
| Attribute masking | 74.7±1.9 | 77.5±3.1 | 59.6±0.7 | 77.9±1.2 | **77.2±0.4** | 78.3±1.1 | 63.3±0.8 | 65.6±0.9 | 71.8 |
| GCC | 74.1±1.4 | 73.2±2.6 | 58.0±0.9 | 75.5±0.8 | 76.6±0.5 | 75.0±1.5 | 63.5±0.4 | 66.9±0.7 | 70.4 |
| GPT-GNN | 75.0±2.5 | 74.9±2.7 | 59.3±0.8 | 77.0±1.7 | 76.1±0.4 | 78.5±0.9 | 63.1±0.5 | 67.5±1.3 | 71.4 |
| Grover | 75.8±1.7 | 76.9±1.9 | 60.7±0.5 | 77.8±1.4 | 76.3±0.6 | 79.5±1.1 | 63.4±0.6 | 68.0±1.5 | 72.3 |
| MGSSL (DFS) | 78.1±1.8 | 79.7±2.2 | 60.5±0.7 | **79.5±1.1** | 76.4±0.4 | **79.7±0.8** | 63.8±0.3 | **70.5±1.1** | 73.5 |
| MGSSL (BFS) | **78.7±1.5** | **80.7±2.1** | **61.8±0.8** | 78.8±1.2 | 76.5±0.3 | 79.1±0.9 | **64.1±0.7** | 69.7±0.9 | **73.7** |

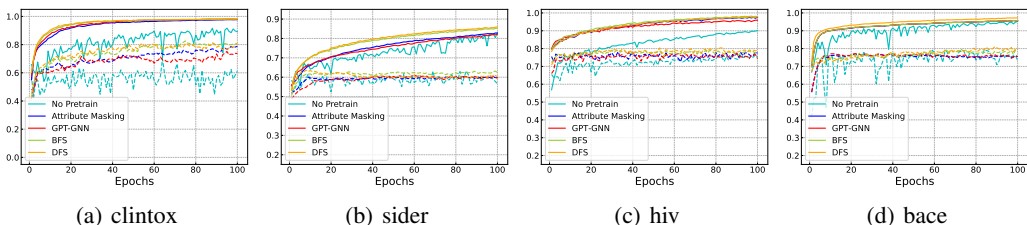

|        (a) clintox        |        (b) sider        |        (c) hiv        |        (d) bace        |

Figure 4: Training and testing curves of different pre-training strategies on GINs. Solid and dashed lines indicate training and testing curves respectively.

In experiments, we also consider GNN without pre-training (direct supervised finetuning) and MGSSL with different generation orders (BFS and DFS).

**Model Configuration.** In the following experiments, we select a five-layer Graph Isomorphism Networks (GINs) as the backbone architecture, which is one of the state-of-the-art GNN methods. Mean pooling is used as the Readout function for GIN. In MGSSL, sum pooling is used to get the embedding of graph motifs. Atom number and chirality tag are input as node features and bond type and direction are regarded as edge features. In the process of pre-training, GNNs are pre-trained for 100 epochs with Adam optimizer and learning rate 0.001. In the finetuning stage, we train for 100 epochs and report the testing score with the best cross-validation performance. The hidden dimension is set to 300 and the batch size is set to 32 for pre-training and finetuning. The split for train/validation/test sets is $80\% : 10\% : 10\%$. All experiments are conducted on Tesla V100 GPUs.

### 4.2 Results and Analysis

**Results on Downstream Tasks.** In Table 1, we show the testing performance on downstream molecular prediction benchmarks using different self-supervised pre-training strategies with GIN. We have the following observations: 1) Generally, GNN models can benefit from various self-supervised pre-training tasks. The average prediction ROC-AUC of all the pre-trained models are better than GNN with direct finetuning. 2) MGSSL methods achieve the best performance on 7 out of 8 benchmarks, demonstrating the effectiveness of motif-based self-supervised pre-training. 3) We also show MGSSL with two motif generation orders in Table 1. Both methods show significant performance improvements on downstream tasks and BFS has a small edge over DFS on average. This may be explained by the fact that MGSSL is required to generate motifs layer-wise in BFS orders, which helps GNNs learn more structural information of motifs. In the following experiments, we use BFS order as the default setting.

In Figure 4, we further show the training and testing curves of MGSSL (BFS and DFS) and the selected baselines. Due to the page limits, we select 4 benchmark datasets here. Beyond predictive performance improvement, our pre-trained GNNs have faster training and testing convergence than baseline methods. Since the pre-training is a one-time-effort, once pre-trained with MGSSL, the pre-trained GNNs can be used for various downstream tasks with minimal finetuning overhead.

**Influence of the Base GNN.** In Table 2 we show that MGSSL is agnostic to the GNN architectures by trying five popular GNN models including GIN [46], GCN [19], RGCN[33], GraphSAGE[11] and DAGNN [24]. We report the average ROC-AUC and the relative gains on 8 benchmarks. We

Table 2: Compare pre-training gains with different GNN architectures, averaged ROC-AUC (%) on 8 benchmark datasets

| Model | GCN | GIN | RGCN | DAGNN | GraphSAGE |
|---|---|---|---|---|---|
| No pretrain | 68.8 | 67.0 | 68.3 | 67.1 | 68.3 |
| MGSSL (BFS) | 72.7 | 73.7 | 73.0 | 72.3 | 73.4 |
| Relative gain | 5.7% | 10.0% | 6.9% | 7.7 % | 7.5% |

observe that all these GNN architectures can benefit from motif-based pre-training tasks. Moreover, GIN achieves the largest relative gain and the best performance after pre-training.

**Influence of Molecule Fragmentation.**
Here we show proper molecule fragmentation methods are vital for the motif-based pre-training. Given different molecule fragmentation methods, different motif vocabulary are generated with varying sizes. Other than the fragmentation method introduced in this paper, we also try other fragmentation schemes with different granularities [17, 4]. BRICS alone [4] tends to generate motifs with large numbers of atoms. Due to the combinatorial explosion, its

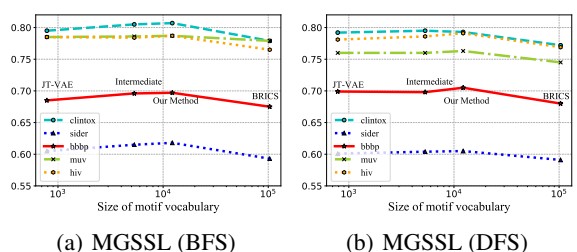

(a) MGSSL (BFS)    (b) MGSSL (DFS)

Figure 5: Influence of the size of motif vocabulary

generated motif vocabulary has a size over 100k while more than 90% motifs have frequencies less than 5. On the other hand, JT-VAE [17] fragments molecules into rings and bonds and has a motif vocabulary size of less than 800. Our methods generate around 12k distinct motifs. By combining different fragmentation strategies, we are able to fragment molecules with intermediate granularities. In Figure 5, we show the influence of the size of motif vocabulary on 5 benchmark datasets. We can observe that the pre-trained models achieve the optimal performance with the motif vocabulary generated by our method. This may be explained by the following reasons: 1) When the motif segmentation is too coarse and the motif vocabulary is too large, the generated motif trees have fewer nodes. It is harder for GNNs to capture the structural information of motifs. Moreover, the generated motifs have low occurrence frequencies, which prevents GNNs from learning the general semantic information of motifs that can be generalized to downstream tasks. 2) When the motif segmentation is too fine, many generated motifs are single atoms or bonds, which inhibits GNNs from learning higher level semantic information through motif generation tasks.

**Ablation Studies on Multi-level Self-supervised Pre-training.** We perform ablation studies to show the effectiveness of the multi-level pre-training. In Table 3, w/o atom-level denotes pre-training GNNs with Motif-level tasks only and the sequential pre-training denotes performing the Motif-level tasks after the Atom-level tasks. As observed from Table 3, the multi-level pre-training has larger average ROC-AUC than the two variants. We can have the following

| Methods | Avg. ROC-AUC |
|---|---|
| w/o atom-level | 73.0 |
| Sequential pre-training | 73.4 |
| Multi-level | **73.7** |

Table 3: Ablation studies on multi-level self-supervised pre-training.

interesting insights: 1) The Atom-level pre-training tasks enables GNNs to first capture the atom-level information, which can benefit higher level, i.e., motif-level tasks. 2) Since our multi-level pre-training unifies multi-scale pre-training tasks and adaptively assigns the weights for hierarchical tasks, it can achieve better performance than sequential pre-training.

## 5 Conclusion and Future Works

In this paper, we proposed Motif-based Graph Self-supervised Learning (MGSSL), which pre-trains GNNs with a novel motif generation task. Through pre-training, MGSSL empowers GNNs to capture the rich semantic and structural information in graph motifs. First, a retrosynthesis-based

algorithm with two additional rules are leveraged to fragment molecule graphs and derive semantic meaningful motifs. Second, a motif generative pre-training framework is designed and two specific generation orders are considered (BFS and DFS). At each step, the pre-trained GNN is required to make topology and motif label predictions. Furthermore, we designed a multi-level pre-training to unify hierarchical self-supervised tasks. Finally, we conducted extensive experiments to show that MGSSL overperforms all the state-of-the-art baselines on various downstream benchmark tasks. Interesting future work includes 1) Designing more self-supervised pre-training tasks based on graph motifs. 2) Exploring motif-based pre-training in other domains other than molecules.

## Acknowledgments and Disclosure of Funding

We thank the anonymous reviewers for valuable feedback. This research was supported by grants from the National Natural Science Foundation of China (Grants No. 61922073 and U20A20229) and 2021 Tencent Rhino-Bird Research Elite Training Program.

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
