# Appendix

## 1 Additional Experiment Settings

In the finetuning process, we consider 8 benchmark binary classification datasets from MoleculeNet. The detailed statistics of datasets are listed in Table 1. The detailed information of these benchmark datasets are listed bellow:

- **muv** is a subset of PubChem BioAssay by applying a refined nearest neighbor analysis. It is designed for the validation of virtual screening techniques.
- **clintox** compares drugs approved through FDA and drugs eliminated due to the toxicity during clinical trials.
- **sider** records marketed drugs along with its adverse drug reactions, also known as the side effect resource.
- **hiv** records the experimentally measured abilities to inhibit HIV replication.
- **tox21** is a public database measuring the toxicity of compounds, which has been used in the 2014 Tox21 Data Challenge.
- **bace** is collected for recording compounds which could act as the inhibitors of (BACE-1) in the past few years.
- **toxcast** contains multiple toxicity labels over thousands of compounds by running high-throughput screening tests on thousands of chemicals.
- **bbbp** involves records of whether a compound carries the permeability property of penetrating the blood-brain barrier.

Table 1: Dataset statistics

| Dataset | muv | clintox | sider | hiv | tox21 | bace | toxcast | bbbp |
|---------|-----|---------|-------|-----|-------|------|---------|------|
| Number of molecules | 93087 | 1478 | 1427 | 41127 | 7831 | 1513 | 8575 | 2039 |
| Number of tasks | 17 | 2 | 27 | 1 | 12 | 1 | 617 | 1 |

## 2 Multi-level Self-supervised Pre-training

Here, we show the pseudo code of the multi-level self-supervised pre-training (Algorithm 1). We adopt the MGDA-UB algorithm from multi-task learning to efficiently solve the optimization problem. Since MGDA-UB calculates the weights $\lambda_i$ by Frank-Wolfe algorithm in each training step, we do not have to provide the weights explicitly.

The input of Multi-level self-supervised pre-training is the molecule datasets, a list of self-supervised tasks. The output is the model parameters of pre-trained GNNs. At the beginning of pre-training, we first randomly initialized the task-specific paramters and the parameters of the pre-trained GNN model. In each training iteration, we update the parameters of all the pretext models. Then we

Submitted to 35th Conference on Neural Information Processing Systems (NeurIPS 2021). Do not distribute.

---
**Algorithm 1** Multi-level Self-supervised Pre-training
---
**Input**: Set of input molecules $X$, set of self-supervised tasks $T$, learning rate $\alpha$
**Output**: Pre-trained parameter $\theta$;
1: $T = \{atom, bond, motif\}$
2: Initialize self-supervised task-specific parameters, $\phi_{atom}$, $\phi_{bond}$ and $\phi_{motif}$.
3: Initialize the parameters of pre-trained GNN model, $\theta$.
4: **while** not convergence **do**
5:  Randomly sample and pre-process input molecules $X$,
   generate self-supervised labels $Y_{atom}, Y_{bond}, Y_{motif}$.
6:  **for** $i \in T$ **do**
7:    $\phi_i \leftarrow \phi_i - \alpha \nabla_{\phi_i} \mathcal{L}_i(X, Y_i, \theta, \phi_i)$
8:  **end for**
9:  $\lambda_{atom}, \lambda_{bond}, \lambda_{motif} \leftarrow$ FRANKWOLFE $(\theta, \phi_{atom}, \phi_{bond}, \phi_{motif})$
10:  $\theta \leftarrow \theta - \sum_{i \in T} \lambda_i \nabla_Z \mathcal{L}_i(X, Y_i, \theta, \phi_i), Z = GNN(X; \theta)$
11: **end while**
---

apply the Frank-Wolfe algorithm [1] to choose the weights, which solves the following optimization problem:

$$\min_{\lambda_i, i \in T} \left\{ \left\| \sum_{i \in T} \lambda_i \nabla_Z \mathcal{L}_i(X, Y_i, \theta, \phi_i) \right\|_2, \quad \sum_{i \in T} \lambda_i = 1, \lambda_i \geq 0 \right\}, \tag{1}$$

where $Z = GNN(X; \theta)$ is the representation of molecules. $\nabla_Z \mathcal{L}_i(X, Y_i, \theta, \phi_i)$ can be computed in a single backward pass for all tasks. With the calculated weights, we can update the parameters of pre-trained GNN model. More details can be found in the original paper of MGDA-UB [2].