# OpenReview forum: "Motif-based Graph Self-Supervised Learning for Molecular Property Prediction"
_NeurIPS.cc/2021/Conference — NeurIPS 2021 Poster_

### Official Review · Reviewer_8tNd · 2021-07-13

**Rating:** 7
**Confidence:** 3

**Summary:**

This paper presents a new self-supervised pre-training framework for Graph Neural Networks (GNNs) called Motif-based Graph Self-supervised Learning (MGSSL) to predict molecular properties. First, the method uses BRICs, a molecule fragmentation method, to create a Graph Motif vocabulary. Next, it performs pre-training using a model with three components -  1) a motif generation module that uses GRUs to recursively generate motifs given the previously generated motifs. 2)  a motif labeling module that uses dense layers followed by a softmax to predict the label of the motif from the vocabulary created in the previous step. 3) a GNN module that generates node and edge embeddings which are finally used to predict node and edge attributes of a molecule graph. MGSSL is trained as a multi-level model (with atom and motif layers) and it minimizes a hybrid loss consisting of Motif, Atomic, and Bond prediction losses, with weights adjusted by the Frank-Wolfe Algorithm. The results report improved performance for molecular property predictions over existing pre-training frameworks 7/8 datasets.


**Limitations And Societal Impact:**

Did not find any specific discussion on the limitations of the work and potential negative societal impacts.

**Main Review:**

Strengths:

++ This paper is a good addition to the recent literature focusing on accurately modeling domain-specific (chemistry) knowledge using deep learning methods. It introduces context-specific mechanisms (incorporating the molecular structure) to provide a performance improvement over established techniques.

++ The different components and levels of the model are interesting and their roles make sense for the task the paper aims to solve.

++ It presents results for 8 different datasets and 6 relevant baselines. It also provides results for different GNN architectures, the influence of the size of motif vocabulary on the performance, and ablation studies to justify the final MGSSL model.

++ The paper places the work with respect to existing methods, lists the contributions, and clearly defines motivations behind them.


Weaknesses:

-- The methodology has multiple and extensive steps, however, the paper does not discuss any limitations related to computational or time cost compared to existing methods. Is the cost/performance tradeoff reasonable?

-- It is unclear whether hyperparameter tuning was performed for all the baseline models when reporting performance. If not, was the choice of default settings fair?

-- In terms of applicability, the MGSSL model seems very specific to the molecular graph domain. Even though the paper presents a good use case for molecular property prediction, its contributions to the wider community are unclear. For example, the BRICS algorithm does not apply to other graph learning tasks. It would be helpful if the paper clarifies its contributions for the wider use of the method.

Minor points:

-An overview figure for the whole model with the components like GRUs and GNNs and connections to the method section would be very useful.

-Figure 4 presents 4 benchmark datasets due to page limits, the rest can be adjusted in the supplementary

Errata: Line 220: “parent motif i generate → generates”



**Time Spent Reviewing:**

2

---

> ### Author Response · Authors · 2021-08-08
> **Response to Reviewer #4**
>
> We thank the reviewer for the detailed and valuable suggestions.
>
> **Comment 1**: The methodology has multiple and extensive steps; however, the paper does not discuss any limitations related to computational or time cost compared to existing methods. Is the cost/performance tradeoff reasonable?
>
> **Response 1**: In our experiments, it took around 20hrs to finish the pretraining on one V100 GPU, which is comparable to other existing methods (e.g., around 24hrs for GPT-GNN [1] in our setting). Since the pre-training is a one-time effort, once pre-trained with MGSSL, the pre-trained GNNs can be used for various downstream tasks with minimal finetuning overhead. Additionally, in Figure 4, we show that using pre-trained GNNs can result in significantly faster convergence compared to GNNs without pre-training, suggesting we can use fewer epochs for downstream task training. Therefore, the cost/performance tradeoff is reasonable.
>
> **Comment 2**: It is unclear whether hyperparameter tuning was performed for all the baseline models when reporting performance. If not, was the choice of default settings fair?
>
> **Response 2**: We selected the hyperparameters based on the baseline papers and then tuned them by grid search.
>
> **Comment 3**: In terms of applicability, the MGSSL model seems very specific to the molecular graph domain. Even though the paper presents a good use case for molecular property prediction, its contributions to the wider community are unclear. For example, the BRICS algorithm does not apply to other graph learning tasks. It would be helpful if the paper clarifies its contributions for the wider use of the method.
>
> **Response 3**: We thank the reviewer for the valuable suggestion. In this work, we focus on the molecular graph domain. Our MGSSL framework is applied for molecular property prediction and BRICS algorithm is leveraged for molecular graph segmentation. According to previous works [2], motif structures widely exist in various graph-structured data such as protein networks. We will explore the application of MGSSL in other graph learning tasks in the future.
>
> **Comment 4**: Minor points related to the overview figure, figure 4, and typos.
>
> **Response 4**: We thank the reviewer for the detailed suggestions. We will update our paper in the future version.
>
> [1] Hu Z, Dong Y, Wang K, et al. Gpt-gnn: Generative pre-training of graph neural networks[C]//Proceedings of the 26th ACM SIGKDD International Conference on Knowledge Discovery & Data Mining. 2020: 1857-1867.
> [2] Milo R, Shen-Orr S, Itzkovitz S, et al. Network motifs: simple building blocks of complex networks[J]. Science, 2002, 298(5594): 824-827.

---

> > ### Comment · Reviewer_8tNd · 2021-08-24
> > **Response to authors' comment**
> >
> > Thank you for your addressing the points raised by the reviewers. After reading other reviews and responses, I'm increasing my rating of the paper.

---

### Official Review · Reviewer_QYy7 · 2021-07-15

**Rating:** 8
**Confidence:** 4

**Summary:**

The authors present a novel pre-training method (Motif-based Graph Self-Supervised Learning, or MGSSL) for graph neural networks specific to molecular property prediction. The pre-training involves constructing synthetic tasks to predict atom information, generation of the molecule, and motif identity. They demonstrate this pre-training improves GNN model performance on several MoleculeNet tasks across 5 different GNN models.

**Limitations And Societal Impact:**

Overall, the authors did a good job of comparing against baselines and analyzing the effect of different components of MGSSL (i.e. pre-training components, motif library creation).

For societal impact, have you considered the computational costs/savings associated with pretraining on MGSSL?

An additional limitation to consider: I would be interested to know how well MGSSL applies to areas of chemical space outside of drug-like molecules. The authors currently use the ZINC dataset, a collection of drug-like molecules, and the fragments are generated using BRICS, which generated fragments for retrosynthesis of drug-like molecules. The [ZINC dataset itself contains biases](https://www.pnas.org/content/116/24/11624); I wonder whether MGSSL also captures these biases.

For applications in other areas of chemical space, such as organic photovolataic molecules, would MGSSL have a lower rate of improvement, or even hurt the model's prediction?

How well do you think MGSSL would work for new areas of drug-like chemical space that haven't been explored before?


**Main Review:**

## Originality
Rating: High

- [+] The use of BRICS to generate motifs for self-supervised pre-training is surprising and novel. The authors compare this approach of generating motifs to the fragments proposed by the JT-VAE and found that using the larger BRICS fragment library was helpful for pre-training the model.


## Quality
Rating: Moderate to High
- [+] The MGSSL approach is applied to 5 different graph neural network models, and compared against the performance of 5 self-supervised pre-training methods as well as GNNs without pretraining
- [+] The authors perform ablation studies to demonstrate the effects of different parts of the pre-training, atom-level and sequential pre-training.
- [+] The authors analyze the impact of the fragmentation method/motif vocabulary size on model improvement after pre-training.
- In Table 1, can you provide an explanation for why the MGSSL pre-training approach does not not show as much improvement over the other SSL methods?

## Clarity
Rating: Moderate to High
- [+] The description of MGSSL pre-training was very clear; I was able to follow the approach taken by the authors.
- For the baseline GNN models, did you train the GNN models for additional epochs? Or were they given the same number of epochs as the pre-trained models? Based on the graphs shown in Figure 4, it seems that the base GNN model (not pretrained) has not yet reached convergence on some of the tasks.
- In table 1, please specify what the values right of the $\pm$ are. I see from 4.1 you mention these are standard deviations of 3 runs of scaffold splitting (I assume?) but it might be helpful to mention this in the table.
- Please include additional architectural information (embedding sizes, full atom/bond feature set) in the supplementary information, if not in the main text.


## Significance
Rating: High

MGSSL improves performance on the selected tasks for a variety of GNN models. Assuming the authors provide code for MGSSL, or the pre-trained model reported in this work, MGSSL could become a very useful starting point for any molecular property prediction project.

I have some questions about the generalizability of MGSSL outside of drug-like chemical space, which I discuss in the Limitations section below.


## Other notes/comments
Section 2 on molecular property prediction, the authors state 'Recently, ... many works regard molecules as graphs and explore the graph convolutional network for property prediction.'

While the use of graph convolutional networks are recent, I would argue that molecules have been regarded as graphs for a long time. Molecular fingerprints such as [ECFP](https://pubs.acs.org/doi/10.1021/ci100050t) and other [cheminformatics descriptors](http://www.talete.mi.it/products/dragon_molecular_descriptors.htm) contain on graph-based features of molecules, such as path length, topological indices. In my view, GNNs provides a way of learn which of these descriptors are most important, but it is still working with the same features.


**Time Spent Reviewing:**

9

---

> ### Author Response · Authors · 2021-08-08
> **Response to Reviewer #3**
>
> We thank the reviewers for their insightful feedback. The reviewer asks great questions, and we provide the answers below.
>
> **Comment 1**: In Table 1, can you provide an explanation for why the MGSSL pre-training approach does not show as much improvement over the other SSL methods?
>
> **Response 1**: According to Table 4, our proposed method MGSSL has an average 6.7% improvement compared with no pretrain. Compared with the strongest pre-train baseline, MGSSL still has an average 1.3% improvement. The limited improvement over other SSL methods may be due to the distribution gap between the pretraining dataset and the finetuning dataset. The fact that some functional groups in the finetuning dataset e.g., C1Nc2ccccc2n2nnnc21 do not exist in the pretraining dataset limits the performance of motif-level self-supervised pretraining.
>
> **Comment 2**: For the baseline GNN models, did you train the GNN models for additional epochs? Or were they given the same number of epochs as the pre-trained models?
>
> **Response 2**: The baseline GNN models are given the same number of epochs as the pre-trained models, which is consistent with previous works [1-2].
>
> **Comment 3**: Please include additional architectural information (embedding sizes, full atom/bond feature set) in the supplementary information
>
> **Response 3**: We thank the reviewer for the valuable suggestion. We will update the supplementary to include additional architectural information.
>
> **Comment 4**:  While the use of graph convolutional networks are recent, I would argue that molecules have been regarded as graphs for a long time. Molecular fingerprints such as ECFP and other cheminformatics descriptors contain on graph-based features of molecules, such as path length, topological indices. In my view, GNNs provides a way of learn which of these descriptors are most important, but it is still working with the same features.
>
> **Response 4**: We thank the reviewer for pointing out the imprecise aspects of our related work. We will update and reorganize the related work in the future version.
>
> **Comment 5**: For societal impact, have you considered the computational costs/savings associated with pretraining on MGSSL?
>
> **Response 5**: In our experiments, it took around 20hrs to finish the pretraining on one V100 GPU, which is comparable to other existing methods. Since the pre-training is a one-time effort, once pre-trained with MGSSL, the pre-trained GNNs can be used for various downstream tasks with minimal finetuning overhead. Additionally, in Figure 4, we show that using pre-trained GNNs can result in significantly faster convergence compared to GNNs without pre-training, suggesting we can use fewer epochs for downstream task training.
>
> **Comment 6**: An additional limitation to consider: I would be interested to know how well MGSSL applies to areas of chemical space outside of drug-like molecules. The authors currently use the ZINC dataset, a collection of drug-like molecules, and the fragments are generated using BRICS, which generated fragments for the retrosynthesis of drug-like molecules. The ZINC dataset itself contains biases; I wonder whether MGSSL also captures these biases.
>
> **Response 6**: In this work, we focus on the molecular graph domain. According to previous works [3], motif structures widely exist in various graph-structured data such as protein networks. We will explore the application of MGSSL in other graph learning tasks in the future. Please also see the response to Comment 3 below for experiments in the organic photovoltaic (OPV) dataset.
>
> Previous work [4] shows that graph neural networks trained on the Zinc dataset may not correctly learn the binding mechanism. In our work, we only pre-train the GNNs on unlabeled molecules from the Zinc dataset. Therefore, there is no bias issue similar to the previous work [4].
>
> **Comment 7**: For applications in other areas of chemical space, such as organic photovoltaic molecules, would MGSSL have a lower rate of improvement, or even hurt the model's prediction?
>
> **Response 7**: We thank the reviewer for the suggestions. We performed additional experiments to show how the property prediction tasks of organic photovoltaic molecules can benefit from MGSSL. We sample 1000 molecules from the (Organic Photovoltaic) OPV datasets [3] for finetuning and predict the HOMO-LUMO gap. The model and training settings are the same as reported in the paper. The MAE of no pre-train model is 0.1772 eV while the MAE of the pre-trained model by MGSSL is 0.1338 eV. The performance improvement shows the effectiveness of MGSSL in other areas of chemical space.
>
> **Comment 8**: How well do you think MGSSL would work for new areas of drug-like chemical space that haven't been explored before?
>
> **Response 8**: We believe that the MGSSL framework is promising for new areas of drug-like chemical space that haven’t been explored before. MGSSL enables GNNs to capture the structural and semantic information of motifs in molecules that can be generalized to unexplored chemical space. We will conduct studies on this as one of our future works.
>
> **References**:
> [1] Hu W, Liu B, Gomes J, et al. Strategies for pre-training graph neural networks[J]. arXiv preprint arXiv:1905.12265, 2019.
> [2] Rong Y, Bian Y, Xu T, et al. Self-supervised graph transformer on large-scale molecular data[J]. arXiv preprint arXiv:2007.02835, 2020.
> [3] Milo R, Shen-Orr S, Itzkovitz S, et al. Network motifs: simple building blocks of complex networks[J]. Science, 2002, 298(5594): 824-827.
> [4] McCloskey K, Taly A, Monti F, et al. Using attribution to decode binding mechanism in neural network models for chemistry[J]. Proceedings of the National Academy of Sciences, 2019, 116(24): 11624-11629.
> [5] Lopez S A, Pyzer-Knapp E O, Simm G N, et al. The Harvard organic photovoltaic dataset[J]. Scientific data, 2016, 3(1): 1-7.

---

> > ### Comment · Reviewer_QYy7 · 2021-08-30
> > **Thanks for the repsonses**
> >
> > Thank you authors for addressing my comments, especially with regards to the applicability of the MGSSL to other chemical spaces.
> >
> > Upon discussion with the other reviewers and the AC, I have decided to lower my rating for two reasons:
> > - As one of the reviewers brought up in additional discussion, this paper would be more complete if additional GNN models were examined  besides GIN.
> > - I had to recalibrate my score upon clarification of the scoring guidelines.

---

> > > ### Author Response · Authors · 2021-08-30
> > > **Response to Reviewer #3**
> > >
> > > We thank the reviewer for the valuable suggestions.
> > >
> > > In Table 2 we tried four additional GNN models with MGSSL and observed the performance gains compared with finetuning from scratch. We admit that comparing MGSSL with other baseline pre-training methods on additional GNN models would make the paper more complete. We will add more complete evaluations in the final version.

---

### Official Review · Reviewer_FWqv · 2021-07-15

**Rating:** 4
**Confidence:** 4

**Summary:**

 This paper proposes Motif-based Graph Self-supervised Learning, which pre-trains GNNs with a novel motif generation task. The authors first use a retrosynthesis-based algorithm with two additional rules to fragment molecule graphs and get meaningful motifs. Next they propose a motif generative pre-training framework, where in each step the pretrained GNN is required to make topology and motif label predictions.

**Ethical Concerns:**

n.a.

**Limitations And Societal Impact:**

The authors do not mention limitations and potential negative societal impact of their work

**Main Review:**

Strengths:
1. The idea of pretraining GNN with motif prediction is interesting;
2. The paper is well-written and easy to follow.

Weaknesses:
1. The authors mentioned that they use BRICS algorithm with two added rules to segment molecule graphs. However, my concern is that if there is a standard definition or criteria on the granularity of functional groups. For example, should C(=O)OH be seen as a whole or two functional groups: C=O and OH? Should C6H5-OH (phenol) be treated as a whole or two functional groups: benzene ring and -OH? Will this granularity affect the performance of the proposed model?
2. BERT can use this pretraining strategy, i.e., predicting the masked word using its context, because adjacent words are seen as related. However, I was wondering if motifs in a molecule also have such property. Given all motifs but one in a molecule graph, can we really predict what the missing motif is? Is there any relationship between adjacent functional groups in a molecule? I suggest that the authors conduct an empirical study on this to show correlation of motifs.
3. According to Table 4, the proposed model does not show significant improvement over baselines.


**Time Spent Reviewing:**

1

---

> ### Author Response · Authors · 2021-08-08
> **Response to Reviewer #2**
>
> We appreciate the constructive comments for our work, which will help us improve our work in the future. Please see below for detailed responses to the comments.
>
> **Comment 1**: The authors mentioned that they use BRICS algorithm with two added rules to segment molecule graphs. However, my concern is that if there is a standard definition or criteria on the granularity of functional groups. For example, should C(=O)OH be seen as a whole or two functional groups: C=O and OH? Should C6H5-OH (phenol) be treated as a whole or two functional groups: benzene ring and -OH? Will this granularity affect the performance of the proposed model?
>
> **Response 1**: We thank the reviewer for this interesting comment. Traditionally in chemistry, C(=O)OH is usually considered as a whole functional group (carboxyl group); for C6H5-OH, OH is usually considered as a group (hydroxyl group). However, as far as we know, these rules are not rigorously defined.
>
> We performed a detailed analysis on the effect of molecular fragmentation in Section 4.2 “Influence of molecular fragmentation”. In figure 5, we show that our model is moderately sensitive to the size of motif vocabulary. A larger vocabulary size indicates more granular motifs and vice versa. Figure 5 also shows that our fragmentation method (BRICS with 2 additional rules) provides the optimal performance.
>
> **Comment 2**: BERT can use this pretraining strategy, i.e., predicting the masked word using its context, because adjacent words are seen as related. However, I was wondering if motifs in a molecule also have such property. Given all motifs but one in a molecule graph, can we really predict what the missing motif is? Is there any relationship between adjacent functional groups in a molecule? I suggest that the authors conduct an empirical study on this to show correlation of motifs.
>
> **Response 2**: We thank the reviewer for the valuable suggestion. Previous works on molecule completion and generation [1-2] showed that capturing the inter-dependence among motifs is vital for generating valid molecules, indicating strong correlation between adjacent functional groups. During the generative pre-training stage, we also observe that some motif pairs often appear together, such as C1=NN2C=NC3=C(C=CO3)C2=N1 and C1CSSCCNN1, CC1=NC2=CC=CC=C2O1 and C1=NN=NC1. Due to the time constraint in the rebuttal period, we believe the systematic study on the correlation of adjacent motifs is best left for future work.
>
> **Comment 3**: According to Table 4, the proposed model does not show significant improvement over baselines.
>
> **Response 3**: According to Table 4, our proposed method MGSSL has an average 6.7% improvement compared with no pretrain. Compared with the strongest pretrain baseline, MGSSL still has an average 1.3% improvement. The limited improvement over other SSL methods may be due to the distribution gap between the pretraining dataset and the finetuning dataset. The fact that some functional groups in the finetuning dataset e.g., C1Nc2ccccc2n2nnnc21does not exist in the pretraining dataset limits the performance of motif-level self-supervised pertaining.
>
> **References**:
> [1] Jin W, Barzilay R, Jaakkola T. Junction tree variational autoencoder for molecular graph generation[C]//International conference on machine learning. PMLR, 2018: 2323-2332.
> [2] Jin W, Barzilay R, Jaakkola T. Hierarchical generation of molecular graphs using structural motifs[C]//International Conference on Machine Learning. PMLR, 2020: 4839-4848.

---

> > ### Comment · Reviewer_FWqv · 2021-08-26
> > **thanks for response**
> >
> > Thanks for the authors' response. I have read the authors' response:
> >
> > Regarding Q1, thanks for providing additional experimental results, which addresses my concern.
> >
> > Regarding Q2, I think this is the key to the proposed method, which could show that the proposed method is well-motivated. Though the authors give some examples that nearby motifs are correlated, it is not enough to show that it is a general rule.
> >
> > Regarding Q3, I searched the experimental results in related work, and I found that the results reported in this paper is significantly lower than the literature. For example, in [1], the best AUC on BBBP is 0.762, while the best AUC reported in this paper is 0.705; the best AUC on BACE is 0.866, while the best AUC reported in this paper is 0.797. In [2], the best AUC on Clintox is 0.906, while in this paper is 0.807. **I think the improvement by introducing the self-supervision task in this work is largely due to the low performance of the the baseline methods (i.e. GIN), which greatly limits the effectiveness of the proposed method.** Therefore, I would like to decrease my score. Thanks.
> >
> > [1] https://arxiv.org/pdf/2011.13230.pdf
> > [2] https://arxiv.org/pdf/2010.09885.pdf

---

> > > ### Author Response · Authors · 2021-08-26
> > > **Response to Reviewer #2**
> > >
> > > We thank the reviewer for the response and questions.
> > >
> > > Regarding Q2, we did empirical studies to count the co-occurrence of motifs and found some examples that nearby motifs are correlated. The focus of our work is the proposed MGSSL framework. Due to time constraint ,we think a systematic study is best left for future work.
> > >
> > > Regarding Q3, the main contribution of our paper is the proposed Motif-based Graph Self-supervised Learning (MGSSL) method, which  is agnostic to base models. **In experiments, we adopt GIN as the backbone model for all the pre-training methods and scaffold splitting for dataset separation, which is consistent with previous works published in ICLR'20 [1]  and ICML'21 [2]. We compare different pre-training methods in Table 1. The experimental results of baseline methods are also consistent with [1] and [2].**
> > >
> > > **Part of our results is lower than that reported in two related works [3, 4] because they use more powerful base models such as transformer.** The focus and settings of our work and [3, 4] are different, which makes the direct comparison unfair. In experiments, we show that MGSSL is agnostic to the GNN architectures by trying five popular GNN models. We will explore using transformer as the backbone model in the future.
> > >
> > > **Moreover, we observe that some results in our paper are higher than results in [3, 4].** For example, the best AUC on hiv in [3] is 0.783 while the best AUC on hiv is 0.795. The best AUC on tox21 in [4] is 0.728 while the best AUC on tox21 in our work is 0.772.
> > >
> > > Finally, the size of pre-training dataset in [3] and [4] (10M and 1.6M respectively) is much larger than our work (0.25 M). It may be not fair to directly compare the results.
> > >
> > > We thank the reviewer for the suggestions and hope our response can address the concern of the reviewer.
> > >
> > > [1] https://arxiv.org/abs/1905.12265
> > > [2] https://arxiv.org/abs/2106.04113
> > > [3] https://arxiv.org/pdf/2011.13230
> > > [4] https://arxiv.org/pdf/2010.09885

---

### Official Review · Reviewer_VNdp · 2021-07-15

**Rating:** 7
**Confidence:** 4

**Summary:**

*Summary

This paper presents a novel motif-level GNN pretraining called "Motif-based Graph Self-Supervised Learning (MGSSL)" and multi-level self-supervised pre-training. Most existing self-supervised pre-training frameworks for GNNs are only defined as node-level or graph-level tasks. But molecular graphs would follow a compositionality principle, and the complex structures are defined by their parts (motifs/fragments/scaffolds) and how to combine them.

It leverages the BRICS fragmentation algorithm [4] and further refinements to decompose a molecule into a "motif tree" representation that describes parts-level compositions of the given molecule in a tree form. Along with atom-level self-supervised tasks by atom-label masking, topology and label prediction for this 'motif tree' defines a nice self-supervised task at a motif level.

Extensive experiments on multiple downstream tasks demonstrated that the proposed pretraining framework outperforms all other state-of-the-art baselines.


**Limitations And Societal Impact:**

Any limitations are not clearly described. It is appreciated to summarize them somewhere in the manuscript?

**Main Review:**

Overall I liked the idea of this paper, and it provides a very interesting approach to GNN pretraining for molecular tasks. Related work is also quite relevant, and extensive experiments for multiple downstream tasks with scaffold-split are very nice to see. Also, it is also nice to see that several common GNN architectures (GCN, GIN, RGCN, DAGNN, GraphSAGE) are fed to the proposed pretraining framework, and observed performance gains for all cases. Comparisons to vocabularies of vanilla BRICS as well as JT-VAE fragments are very informative, too.

The following are some points unclear to me, and it will be nice to be in a clear description.

1.
GNN models are described just as function f_theta, but this symbol is not used explicitly in the main paper. Because of this, it is a bit unclear what is assumed as the input to GNNs, and how we can use the pretrained GNNs for downstream tasks such as property prediction.

We have two graphs, input molecular graph G + the corresponding 'motif tree' T(G). Are GNNs pretrained as a function G -> vector? Or T(G) to vector, and when we use it in downstream tasks, T(G) needs to be constructed, keeping the same BRICS+alpha vocabulary to label T(G) accordingly??

Furthermore, multi-level cases, do we need two GNNs pretrained for both atom-level and motif-level? Also, use both two GNNs for transfer to downstream tasks?

It is appreciated to clearly describe the input-output of GNNS using the defined symbol f_theta, how to use the output vector for pretraining, and how to use the pretrained GNNs for downstream tasks.

2.
Similarly, line 196 says "the question is ... how to model the conditional probability log p_theta(..) ", but this probability is never explicitly defined. So this implies minimizing the loss eq.(13) corresponds to maximizing the log likelihood (the conditional probability?). Any explicit and consistent descriptions are appreciated.


**Time Spent Reviewing:**

4 hours

---

> ### Author Response · Authors · 2021-08-08
> **Response to Reviewer #1**
>
> We thank the reviewer for the appreciation and valuable comments. As for unclear parts in the paper, we will add more clarifications and details in the future version.
>
> **Comment 1**: GNN models are described just as function f_theta, but this symbol is not used explicitly in the main paper. Because of this, it is a bit unclear what is assumed as the input to GNNs, and how we can use the pretrained GNNs for downstream tasks such as property prediction.
> We have two graphs, input molecular graph G + the corresponding 'motif tree' T(G). Are GNNs pretrained as a function G -> vector? Or T(G) to vector, and when we use it in downstream tasks, T(G) needs to be constructed, keeping the same BRICS+alpha vocabulary to label T(G) accordingly??
> Furthermore, multi-level cases, do we need two GNNs pretrained for both atom-level and motif-level? Also, use both two GNNs for transfer to downstream tasks?
> It is appreciated to clearly describe the input-output of GNNS using the defined symbol f_theta, how to use the output vector for pretraining, and how to use the pretrained GNNs for downstream tasks.
>
> **Response 1**:
> The inputs to the GNNs are the attributed molecule graphs, including the node features, edge features, and the adjacency matrix. The pre-trained GNNs can output better representations of input molecules. For specific downstream tasks, we can finetune the pre-trained model and achieve better performance than training from scratch.
>
> In MGSSL, the GNNs are pre-trained as a function G-> vector. T(G) is only constructed and used in the motif-level self-supervised tasks and is not required in downstream tasks.
>
> For the multi-level cases, we only need one GNN as the graph encoder. We use several prediction heads for multi-level self-supervised tasks to take the multi-scale information in molecular graphs into consideration. After the pretraining, the prediction heads are removed in the downstream tasks.
>
> **Comment 2**: Similarly, line 196 says "the question is ... how to model the conditional probability log p_theta(..) ", but this probability is never explicitly defined. So this implies minimizing the loss eq.(13) corresponds to maximizing the log-likelihood (the conditional probability?). Any explicit and consistent descriptions are appreciated.
>
> **Response 2**: We thank the reviewer for the valuable suggestion. Yes, maximizing the log-likelihood corresponds to minimizing the optimization objective in eq.(13) in our paper.
>
> **Comment 3**: Any limitations are not clearly described. It is appreciated to summarize them somewhere in the manuscript?
>
> **Response 3**: We thank the reviewer for the constructive suggestion. We will add discussions on limitations and societal impact in the future version. As pointed by other reviewers, one limitation may be the computational cost of pretraining. However, the pre-training is a one-time effort. Once pre-trained with MGSSL, the pre-trained GNNs can be used for various downstream tasks with minimal finetuning overhead and faster convergence speed. The cost/performance tradeoff is still reasonable.

---

> > ### Comment · Reviewer_VNdp · 2021-08-25
> > **Thank you for the response**
> >
> > Thank you for the response. It was nice to make sure that $T(G)$ is only constructed and used only in the motif-level SSL tasks!
> >
> > So does this mean that $T(G)$ is like $T(f_\theta(G))$, and $T$ is only needed to define the motif-level SSL objective?? Also, the embedding of motif $i$, $x_i$, at line 213 would be actually the output from GNN $f_\theta$ like $x_i = f_\theta(G_i)$ where $G_i$ is the molecular graph of the motif $i$? (or something like $x_i = \mathrm{readout}(f_\theta(G_i))$)
> >
> > This point was not clear to me from the descriptions at line 175-179, and it'll be nice to see that the unused symbol $f_\theta$ is somehow explicitly used in the model and SSL-objective descriptions in the final version!

---

> > > ### Author Response · Authors · 2021-08-25
> > > **Response to Reviewer #1**
> > >
> > > We thank the reviewer for the response and valuable suggestions.
> > >
> > > **Question 1:** So does this mean that $T(G)$ is like $T(f_\theta(G))$, and $T$ is only needed to define the motif-level SSL objective?
> > >
> > > **Response 1:** Yes, $T$ is only needed to define the motif-level SSL objective.
> > >
> > > **Question 2:**Also, the embedding of motif $i$, $x_i$, at line 213 would be actually the output from GNN $f_\theta$ like $x_i = f_\theta(G_i)$ where $G_i$ is the molecular graph of the motif $i$? (or something like $x_i = \mathrm{readout}(f_\theta(G_i))$)
> > >
> > > **Response 2:** The embedding of motif i can be obtained by pooling the atom embeddings in motif i, e.g., average pooling. Formally, it can be written as $x_i = \mathrm{readout}(f_\theta(G_i))$.
> > >
> > > We thank the reviewer for the suggestion to explicitly use $f_{\theta}$ in the model and SSL-objective descriptions. We will modify and improve this part of the paper in the final version.

---

### Decision · Program_Chairs · 2021-09-27

**Decision:**

Accept (Poster)

**Comment:**

This paper presents a novel motif-level GNN pretraining. While most existing self-supervised pre-training frameworks for GNNs are only defined as node-level or graph-level tasks, molecular graphs would follow a compositionality principle, and the complex structures are defined by their parts (motifs/fragments/scaffolds) and how to combine them. The authors show that by leveraging the BRICS fragmentation algorithm to decompose a molecule into a "motif tree" representation, the pretraining could become more effective.
This paper triggered a lot of discussions between the authors and the reviewers, as well as among the reviewers themselves. One reviewer raised very serious concerns on the technical novelty (whether leveraging an existing algorithm for molecule decomposition could provide sufficient novelty to the pretraining task), and on the choice of GIN as the backbone model in the experiments (many stronger GNN models were developed recently). On the other hand, other reviewers believed that the introduction of motif to molecule graph learning was novel by its own, especially for this inter discipline, and the use of GIN makes sense and does not affect the main conclusion. Although a consensus was not successfully reached at the end, I tend to agree that this paper has its value in representation of molecule graph (and shed some lights on AI for molecular science), and there is no serious issue with the choice of GIN. Therefore, my recommendation is ACCEPT as a poster.